# NEURAL POTTS MODEL

## ABSTRACT

We propose the Neural Potts Model objective as an amortized optimization problem. The objective enables training a single model with shared parameters to explicitly model energy landscapes across multiple protein families. Given a protein sequence as input, the model is trained to predict a pairwise coupling matrix for a Potts model energy function describing the local evolutionary landscape of the sequence. Couplings can be predicted for novel sequences. A controlled ablation experiment assessing unsupervised contact prediction on sets of related protein families finds a gain from amortization for low-depth multiple sequence alignments; the result is then confirmed on a database with broad coverage of protein sequences.

## 1    INTRODUCTION

When two positions in a protein sequence are in spatial contact in the folded three-dimensional structure of the protein, evolution is not free to choose the amino acid at each position independently. This means that the positions co-evolve: when the amino acid at one position varies, the assignment at the contacting site may vary with it. A multiple sequence alignment (MSA) summarizes evolutionary variation by collecting a group of diverse but evolutionarily related sequences. Patterns of variation, including co-evolution, can be observed in the MSA. These patterns are in turn associated with the structure and function of the protein (Göbel et al., 1994). Unsupervised contact prediction aims to detect co-evolutionary patterns in the statistics of the MSA and infer structure from them.

The standard method for unsupervised contact prediction fits a Potts model energy function to the MSA (Lapedes et al., 1999; Thomas et al., 2008; Weigt et al., 2009). Various approximations are used in practice including mean field (Morcos et al., 2011), sparse inverse covariance estimation (Jones et al., 2011), and pseudolikelihood maximization (Balakrishnan et al., 2011; Ekeberg et al., 2013; Kamisetty et al., 2013). To construct the MSA for a given input sequence, a similarity query is performed across a large database to identify related sequences, which are then aligned to each other. Fitting the Potts model to the set of sequences identifies statistical couplings between different sites in the protein, which can be used to infer contacts in the structure (Weigt et al., 2009). Contact prediction performance depends on the depth of the MSA and is reduced when few related sequences are available to fit the model.

In this work we consider fitting many models across many families simultaneously with parameter sharing across all the families. We introduce this formally as the Neural Potts Model (NPM) objective. The objective is an amortized optimization problem across sequence families. A Transformer model is trained to predict the parameters of a Potts model energy function defined by the MSA of each input sequence. This approach builds on the ideas in the emerging field of protein language models (Alley et al., 2019; Rives et al., 2019; Heinzinger et al., 2019), which proposes to fit a single model with unsupervised learning across many evolutionarily diverse protein sequences. We extend this core idea to train a model to output an explicit energy landscape for every sequence.

To evaluate the approach, we focus on the problem setting of unsupervised contact prediction for proteins with low-depth MSAs. Unsupervised structure learning with Potts models performs poorly when few related sequences are available (Jones et al., 2011; Kamisetty et al., 2013; Moult et al., 2016). Since larger protein families are likely to have structures available, the proteins of greatest interest for unsupervised structure prediction are likely to have lower depth MSAs (Tetchner et al., 2014). This is especially a problem for higher organisms, where there are fewer related genomes (Tetchner et al., 2014). The hope is that for low-depth MSAs, the parameter sharing in the neural model will improve results relative to fitting an independent Potts model to each family.

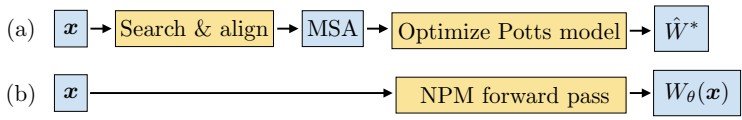

Figure 1: (a) Standard Potts model requires constructing an MSA and optimizing parameters $W$. (b) Neural Potts Model (NPM) predicts $W$ in a single feedforward pass from a single sequence.

We investigate the NPM objective in a controlled ablation experiment on a group of related protein families in PFAM (Finn et al., 2016). In this artificial setting, information can be generalized by the pre-trained shared parameters to improve unsupervised contact prediction on a subset of the MSAs that have been artificially truncated to reduce their number of sequences. We then study the model in the setting of a large dataset without artificial reduction, training the model on MSAs for UniRef50 sequences. In this setting there is also an improvement on average for low depth MSAs both for sequences in the training set as well as for sequences not in the training set.

## 2  BACKGROUND

**Multiple sequence alignments**  An MSA is a set of aligned protein sequences that are evolutionarily related. MSAs are constructed by retrieving related sequences from a sequence database and aligning the returned sequences using a heuristic. An MSA can be viewed as a matrix where each row is a sequence, and columns contain aligned positions after removing insertions and replacing deletions with gap characters.

**Potts model**  The generalized Potts model defines a Gibbs distribution over a protein sequence $(x_1, \ldots, x_L)$ of length $L$ with the negative energy function:

$$-E(\boldsymbol{x}) = \sum_i h_i(x_i) + \sum_{ij} J_{ij}(x_i, x_j) \tag{1}$$

Which defines potentials $h_i$ for each position in the sequence, and couplings $J_{ij}$ for every pair of positions. The parameters of the model are $W = \{h, J\}$ the set of fields and couplings respectively. The distribution $p(\boldsymbol{x}; W)$ is obtained by normalization as $\exp\{-E(\boldsymbol{x}; W)\}/Z(W)$.

Since the normalization constant is intractable, pseudolikelihood is commonly used to fit the parameters (Balakrishnan et al., 2011; Ekeberg et al., 2013). Pseudolikelihood approximates the likelihood of a sequence $\boldsymbol{x}$ as a product of conditional distributions: $\ell_{\mathrm{PL}}(\boldsymbol{x}; W) = -\sum_i \log p(x_i | x_{-i}; W)$. To estimate the Potts model, we take the expectation:

$$\mathcal{L}_{\mathrm{PL}}(W) = \mathop{\mathbb{E}}_{\boldsymbol{x} \sim \mathcal{M}} [\ell_{\mathrm{PL}}(\boldsymbol{x}; W)] \tag{2}$$

over an MSA $\mathcal{M}$. In practice, we have a finite set of sequences $\hat{\mathcal{M}}$ in the MSA to estimate Eq. (2). $L_2$ regularization $\rho(W) = \lambda_J \|J\|^2 + \lambda_h \|h\|^2$ is added, and sequences are reweighted to account for redundancy (Morcos et al., 2011). We write the regularized finite sample estimator as:

$$\hat{\mathcal{L}}_{\mathrm{PL}}(W) = \frac{1}{M_{\mathrm{eff}}} \sum_{m=1}^{M} w^m [\ell_{\mathrm{PL}}(\boldsymbol{x}^m; W)] + \rho(W) \tag{3}$$

Which sums over all the $M$ sequences of the finite MSA $\hat{\mathcal{M}}$, weighted with $w^m$ summing collectively to $M_{\mathrm{eff}}$. The finite sample estimate of the parameters $\hat{W}^*$ is obtained by minimizing $\hat{\mathcal{L}}_{\mathrm{PL}}$.

**Idealized MSA**  Notice how in Eq. (2), we idealized the MSA $\mathcal{M}$ as a distribution, defined by the protein family. We consider the set of sequences actually retrieved in the MSA $\hat{\mathcal{M}}$ in Eq. (3) as a finite sample from this underlying idealized distribution. For some protein families this sample will contain more information than for others, depending on what sequences are present in the database. We will refer to $W^*$ as a hypothetical idealized estimate of the parameters to explain how the Neural Potts Model can improve on the finite sample estimate $\hat{W}^*$ for low-depth MSAs.

## 2.1 Amortized optimization

We review amortized optimization (Shu, 2017), a generalization of amortized variational inference (Kingma & Welling, 2013; Rezende et al., 2014) that uses learning to predict the solution to continuous optimization problems to make the computation more tractable and potentially generalize across problem instances. We are interested in repeatedly solving expensive optimization problems

$$W^*(x) = \arg\min_W \mathcal{L}(W; x), \tag{4}$$

where $W \in \mathbb{R}^m$ is the optimization variable, $x \in \mathbb{R}^n$ is the input or conditioning variable to the optimization problem, and $\mathcal{L} : \mathbb{R}^m \times \mathbb{R}^n \to \mathbb{R}$ is the objective. We assume $W^*(x)$ is unique. We consider the setting of having a distribution over optimization problems with inputs $x \sim p(x)$, and the $\arg\min$ of those optimization problems $W^*(x)$.

Amortization uses learning to leverage the shared structure present across the distribution, *e.g.* a solution $W^*(x)$ is likely correlated with another solution $W^*(x')$. Assuming an underlying regularity of the data and loss $\mathcal{L}$, we can imagine learning to predict the outcome of the optimization problem with an expressive model $W_\theta(x)$ such that hopefully $W_\theta \approx W^*$. Modeling and learning $W_\theta(x)$ are the key design decisions when using amortization.

**Modeling approaches.** In this paper we consider models $W_\theta(x)$ that directly predict the solution to Eq. (4) with a neural network, an approach which follows fully amortized variational inference models and the meta-learning method (Mishra et al., 2017). The model can also leverage the objective information $\mathcal{L}(W; x)$ and gradient information $\nabla_W \mathcal{L}(W; x)$, *e.g.* by predicting multiple candidate solutions $W$ and selecting the most optimal one. This is sometimes referred to as semi-amortization or unrolled optimization-based models and is considered in Gregor & LeCun (2010) for sparse coding, Li & Malik (2016); Andrychowicz et al. (2016); Finn et al. (2017) for meta-learning, and Marino et al. (2018); Kim et al. (2018) for posterior optimization.

**Learning approaches.** There are two main classes of learning approaches for amortization:

$$\arg\min_\theta \mathbb{E}_{p(x)} \mathcal{L}(W_\theta(x); x) \tag{5} \qquad\qquad \arg\min_\theta \mathbb{E}_{p(x)} \|W_\theta(x) - W^*(x)\|_2^2. \tag{6}$$

*Gradient-based* approaches leverage gradient information of the objective $\mathcal{L}$ and optimize Eq. (5) whereas *regression-based* approaches optimize a distance to ground-truth solutions $W^*$, such as the squared $\mathrm{L}^2$ distance in Eq. (6). Prior work has shown that models trained with these objectives can learn to predict the optimal $W^*$ directly as a function of $x$. Given enough regularity of the domain, if we observe new (test) samples $x' \sim p(x)$ we expect the model to generalize and predict the solution to the original optimization problem Eq. (4). Gradient-based approaches have the computational advantage of not requiring the expensive ground-truth solution $W^*$ while regression-based approaches are less susceptible to poor local optima in $\mathcal{L}$. Gradient-based approaches are used in variational inference (Kingma & Welling, 2013), style transfer (Chen & Schmidt, 2016), meta learning (Finn et al., 2017; Mishra et al., 2017), and reinforcement learning, *e.g.* for the policy update in model-free actor-critic methods (Sutton & Barto, 2018). Regression-based approaches are more common in control for behavioral cloning and imitation learning (Duriez et al., 2017; Ratliff et al., 2007; Bain & Sammut, 1995).

## 3 Neural Potts Model

In Eq. (2) we introduced the Potts model for a single MSA $\mathcal{M}$ (aligned set of sequences $\boldsymbol{x}$), to optimize $W^* = \{h^*, J^*\} = \arg\min_W \mathbb{E}_{\tilde{\boldsymbol{x}} \sim \mathcal{M}}[\ell_{\mathrm{PL}}(\tilde{\boldsymbol{x}}; W)]$. As per Eq. (5) We will now introduce a neural network to estimate Potts model parameters from a single sequence: $\{h_\theta(\boldsymbol{x}), J_\theta(\boldsymbol{x})\} = W_\theta(\boldsymbol{x})$ with a single forward pass.

We propose minimizing the following objective for the NPM parameters $\theta$, which directly minimizes the Potts model losses in expectation over our data distribution $\boldsymbol{x} \sim \mathcal{D}$ and their MSAs $\tilde{\boldsymbol{x}} \sim \mathcal{M}(\boldsymbol{x})$:

$$\mathcal{L}_{\mathrm{NPM}}(\theta) = \mathbb{E}_{\boldsymbol{x} \sim \mathcal{D}} \left[ \mathbb{E}_{\tilde{\boldsymbol{x}} \sim \mathcal{M}(\boldsymbol{x})} \ell_{\mathrm{PL}}(\tilde{\boldsymbol{x}}; W_\theta(\boldsymbol{x})) \right] \tag{7}$$

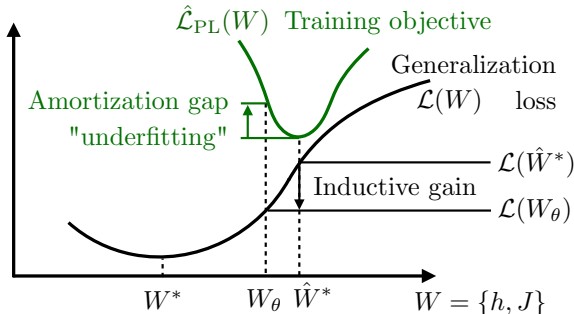

Figure 2: Inductive generalization gain (illustration with a 1D loss landscape). $\hat{W}^*$ is the standard Potts model, estimated on the finite observed MSA $\hat{\mathcal{M}}$. Though it minimizes the training objective, it does not achieve perfect generalization performance. However the Neural Potts Model $W_\theta(\boldsymbol{x})$ can generalize better than $\hat{W}^*$ through transfer learning from related samples, guided by the inductive bias of the model. We expect this especially when the estimate $\hat{W}^*$ is far from $W^*$, e.g. on small or biased MSAs.

To compute the loss for a given sequence $\boldsymbol{x}$ we compute the Potts model parameters $W_\theta(\boldsymbol{x})$, and evaluate its pseudo-likelihood loss $\ell_{\mathrm{PL}}$ on a set of sequences $\tilde{\boldsymbol{x}}$ from the MSA constructed with $\boldsymbol{x}$ as query sequence. This fits exactly in "amortized optimization" in Section 2.1 Eq. (5): we train a model to predict the outcome of a set of highly related optimization problems. One key extension to the described amortized optimization setup is that the model $W_\theta$ estimates the Potts Model parameters from only the MSA query sequence $\boldsymbol{x}$ as input rather than the full MSA $\mathcal{M}(\boldsymbol{x})$. Thus, our model must learn to distill the protein energy landscape into its parameters, since it cannot look up related proteins during runtime. A full algorithm is given in Appendix A.

Similar to the original Potts model, we need to add a regularization penalty $\rho(W)$ to the main objective. For a finite sample of N different query sequences $\{\boldsymbol{x}_n\}$, and a corresponding sample of N× M aligned sequences $\{\tilde{\boldsymbol{x}}_n^m\}$ from MSA $\hat{\mathcal{M}}(\boldsymbol{x}_n)$, the finite sample regularized loss, i.e. NPM training objective, becomes:

$$\hat{\mathcal{L}}_{\mathrm{NPM}}(\theta) = \sum_{n=1}^{N} \left[ \frac{1}{M_{\mathrm{eff}}(n)} \sum_{m=1}^{M} w_n^m \left[ \ell_{\mathrm{PL}}(\tilde{\boldsymbol{x}}_n^m; W_\theta(\boldsymbol{x}_n)) \right] + \rho(W_\theta(\boldsymbol{x}_n)) \right] \tag{8}$$

**Inductive generalization gain** (see Fig. 2) is when the Neural Potts Model improves over the individual Potts model. Intuitively this is possible because the individual Potts Models are not perfect estimates (finite/biased MSAs), while the shared parameters of $W_\theta$ can transfer information between related protein families and from pre-training with another objective like masked language modeling (MLM).

Let us start with the normal amortized optimization setting, where we expect an amortization gap (Cremer et al., 2018). The amortization gap means that $W_\theta(x)$ will be behind the optimal $W^*$ for the objective $\mathcal{L}$: $\mathcal{L}(W_\theta(x)) > \mathcal{L}(W^*)$. This is closely related to underfitting: the model $W_\theta$ is not flexible enough to capture $W^*(x)$. However, recall that in the Potts model setting, there is a finite-sample training objective $\hat{\mathcal{L}}$ (Eq. (8)), with minimizer $\hat{W}^*$. We can expect an amortization gap in the training objective; however this amortization gap can now be advantageous. Even if the amortized solution $W_\theta(\boldsymbol{x})$ is near-optimal on $\hat{\mathcal{L}}$, it can likely find a more generalizable region of the overparametrized domain $W$ by parameter sharing of $\theta$, allowing it to transfer information between related instances. The inductive bias of $W_\theta(\boldsymbol{x})$ can allow the neural amortized estimate to generalize better, especially when the finite sample $\hat{\mathcal{M}}$ is poor. This inductive bias depends on the choice of model class for $W_\theta$, its pre-training, as well as the shared structure between the protein families in the dataset. Concretely we will consider for the generalization loss $\mathcal{L}$ not just the pseudo-likelihood loss on test MSA sequences, but also the performance on downstream validation objectives like predicting contacts, a proxy for the model's ability to capture the underlying structure of the protein.

We will show that for some samples $\mathcal{L}(W_\theta(\boldsymbol{x})) < \mathcal{L}(\hat{W}^*)$, i.e. there is an inductive generalization gain. This is visually represented in Fig. 2; and Table 1 compares amortized optimization and NPM, making a connection to multi-task learning (Caruana, 1998). Additionally, we could frame NPM as a hypernetwork, a neural network that predicts the weights of second network (in this case the Potts model) as in, *e.g.*, Gomez & Schmidhuber (2005); Ha et al. (2016); Bertinetto et al. (2016).

Table 1: Comparison between (A) "standard" amortized optimization, (B) Neural Potts Model, and (C) Multi-task learning. From row (A) amortized optimization to (B) Neural Potts Model, a finite-sample training loss is introduced which comes with considerations of generalization and regularization. This is related to multi-task learning, but with a major difference that (B) the solo optimization is over a single tensor $W$ in the Potts model, but (C) a function $f_\theta$ in a learning problem. In the amortized/multi-task setting, the distribution over query sequences $\boldsymbol{x}$ in (B) NPM plays the role that different related tasks play in (C) MTL. In the NPM setting (B), $W_\theta$ takes $\boldsymbol{x}$ explicitly as argument, versus (C) MTL typically just has a separate output head per task.

| | Solo objective Training | Solo objective Generalization | Amortized / Multi-task | Parametrization + model choices |
|---|---|---|---|---|
| **(A)** Optim$\rightarrow$ Amortized | $\mathcal{L}(s; W)$ | $\mathcal{L}(s; W)$ (= Training) | Amortized optim: $\mathbb{E}_{p(s)}\, \mathcal{L}(s; W_\theta(s))$ | **Solo**: $W \in \mathbb{R}^n$ **Amor**: $W_\theta : \mathbb{R}^d \to \mathbb{R}^n$ +learner class |
| **(B)** Potts $\rightarrow$ NPM | PLL, finite MSA $\hat{M}$: $\hat{\mathcal{L}}(W) = \sum_m \ell_{\mathrm{PL}}(\tilde{\boldsymbol{x}}^m; W)$ | Distr $\mathcal{L}(W) = \mathbb{E}[\ell_{\mathrm{PL}}(\tilde{\boldsymbol{x}}; W)]$ or Contact pred | Neural Potts $\mathbb{E}_{\boldsymbol{x}}[\hat{\mathcal{L}}^{\hat{M}}(W_\theta(\boldsymbol{x}))]$ | **Solo**: $W \in \mathbb{R}^n$ +regularization **Amor**: $W_\theta : \mathbb{R}^d \to \mathbb{R}^n$ +learner class |
| **(C)** ML $\rightarrow$ MTL (Multi-task learning) | ERM: $\hat{\mathcal{L}}(f_\theta) = \sum_m \ell(f_\theta(x_m), y_m)$ | $\mathcal{L}(f_\theta) = \mathbb{E}_{xy} \ell(f_\theta(x), y)$ | Multi-task learning: $\sum_{t=1}^T [\hat{\mathcal{L}}^t(f_\theta^t)]$ for $T$ related tasks | **Solo**: $f_\theta : \mathbb{R}^d \to \mathbb{R}$ +regularization +learner class **MTL**: $f_\theta^t : \mathbb{R}^d \to \mathbb{R}$ + param sharing $f_\theta^t$ |

In summary, the goal for the NPM is to "distill" an ensemble of Potts models into a single feedforward model. From a self-supervised learning perspective, rather than supervising the model with the input directly, we use supervision from an energy landscape around the input.

## 4 EXPERIMENTS

In Section 4.1 we present results on a small set of related protein domain families from Pfam, where we artificially reduce the MSA depth for a few families to study the inductive generalization gain from the shared parameters. In Section 4.2 we present results on a large Transformer trained on MSAs for all of UniRef50.

For the main representation $g_\theta(\boldsymbol{x})$ we use a bidirectional transformer model (Vaswani et al., 2017). To compute the four-dimensional pairwise coupling tensor $J_\theta(\boldsymbol{x})$ from sequence embedding $g_\theta(\boldsymbol{x})$ we introduce the multi-head bilinear form (mhbf) in Appendix B. One can think of the multi-head bilinear form as the $L \times L$ self-attention maps of the Transformer's multi-head attention module, but without softmax normalization. When using mhbf for direct prediction, there are $K^2$ heads, one for every amino acid pair $k, l$. For the Pfam experiments, we extend the architecture with convolutional layers after the mhbf, where the final convolutional layer has $K^2$ output channels. We initialize $g_\theta(\boldsymbol{x})$ with a Transformer pre-trained with masked language modeling following (Rives et al., 2019).

To evaluate Neural Potts Model energy landscapes, we will focus on proteins with structure in the Protein Data Bank (PDB), using the magnitude of the couplings after APC correction to rank contacts. The protocol is described in Appendix C.2.

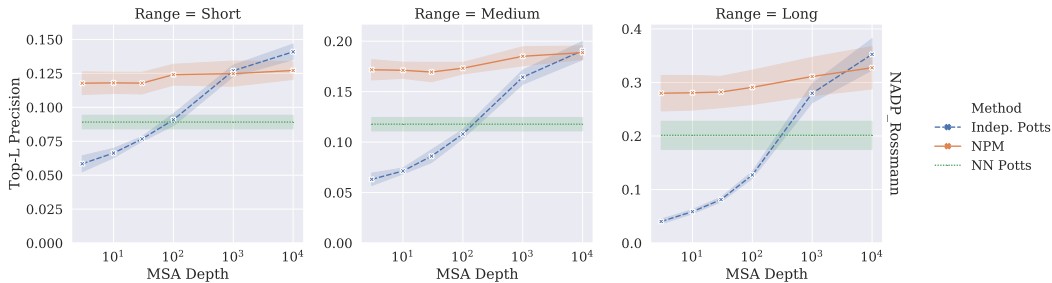

Figure 3: Contact prediction precision on Pfam families from the NADP Rossmann clan, at different levels of depth reduction. Columns show (from left to right) short, medium and long-range precision for top-L threshold. Across the metrics, NPM outperforms the independent Potts model trained on the shallowest MSAs, as well as the Nearest Neighbor Potts model baseline.

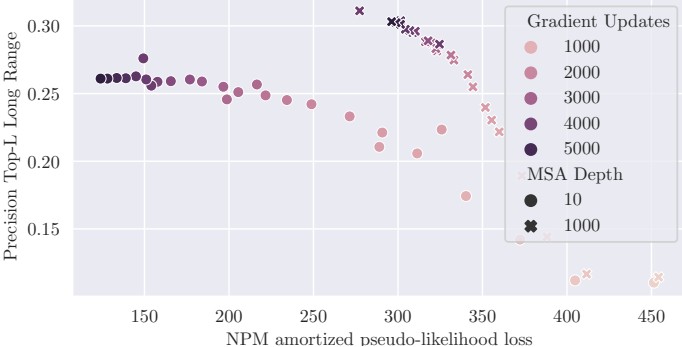

Figure 4: Trajectory of training on the NADP Rossman clan, averaged over five-fold cross-evaluation.

### 4.1 PFAM CLANS

To study generalization in a controlled setting, we investigate a small set of structurally-related MSAs from the Pfam domain family database (Finn et al., 2016) belonging to the same Pfam clan. We expect that on a collection of related MSAs, information could be generalized to improve performance on low-depth MSAs. Families within a Pfam clan are linked by a distant evolutionary relationship, giving them related but not trivially-similar structure. We obtain contact maps for the sequences in each of the families where a structure is available in the PDB. At test time we input the sequence and compare the generated couplings under the model to the corresponding structure.

We compare the NPM to two baselines. The first direct comparison is to an independent Potts model trained directly on the MSA. For the second baseline we construct the "nearest neighbor" Potts model, by aligning each test sequence against all families in the training set, and using the Potts model from the closest matching family.

We perform the experiment using a five-fold cross-evaluation scheme, in which we partition the clan's families into five equally-sized buckets. As in standard cross-validation, each bucket will eventually serve as an evaluation set. However, we do not remove the evaluation bucket. Instead, we artificially reduce the number of sequences in the MSAs in the evaluation bucket to a smaller fixed MSA depth. MSAs in the remaining buckets remain unaltered. The goal of this setup is to check the model's ability to infer contacts on artificially limited sets of sequences. Both NPM and the baseline independent Potts model are fit on the reduced set of sequences. Note that while the baseline Potts model uses the reduced MSA of the target directly, NPM is trained on the reduced MSA but evaluated using only the target sequence as input. We train a separate NPM on each of the five cross-evaluation rounds, evaluate on the structures corresponding to the bucket with reduced

MSAs, and show averages and standard deviations across rounds. Further details are provided for model training in Appendix C.1 and for the Pfam dataset in Appendix C.3.

Figure 3 shows the resulting contact prediction performance on the 181 families in the NADP Rossmann clan, with additional results on the P-loop NTPase, HTH, and AB hydrolase clans in Appendix D Fig. 9. We initialize a 12-layer Transformer with protein language modeling pre-training. Because of the small dataset size, we keep the weights of the base Transformer $g_\theta$ frozen and only finetune the final layers. As a function of increasing MSA depth, contact precision improves for both NPM and independent Potts models. For the shallowest MSAs, NPM has a higher precision relative to the independent Potts models. The advantage at low MSA depth is most pronounced for long range contacts, outperforming independent Potts models up to MSA depth 1000. These experiments suggest NPM is able to realize an inductive gain by sharing parameters in the pre-trained base model as well as the fine-tuned final layers and output head. Figure 4 shows training trajectories. We observe near-monotonic decrease of the amortized pseudo-likelihood loss (Eq. (7)) on the MSAs in the evaluation set, and increase of the top-L long range contact precision. This indicates that improving the NPM objective improves the unsupervised contact precision across the reduced-depth MSAs. Furthermore we see expected overfitting for smaller MSA depth: better training loss but worse contact precision.

Additionally, we assess performance of different architecture variants: direct prediction with the multi-head bilinear form (always using symmetry), with or without tied projections, and addition of convolutional layers after the multi-head bilinear form. The variants are described in detail in Appendix B. We find in Appendix D Fig. 8 that addition of convolutional layers after the multi-head bilinear form performs best; for the variant without convolutional layers, the head without weight tying performs best.

## 4.2 UNIREF50

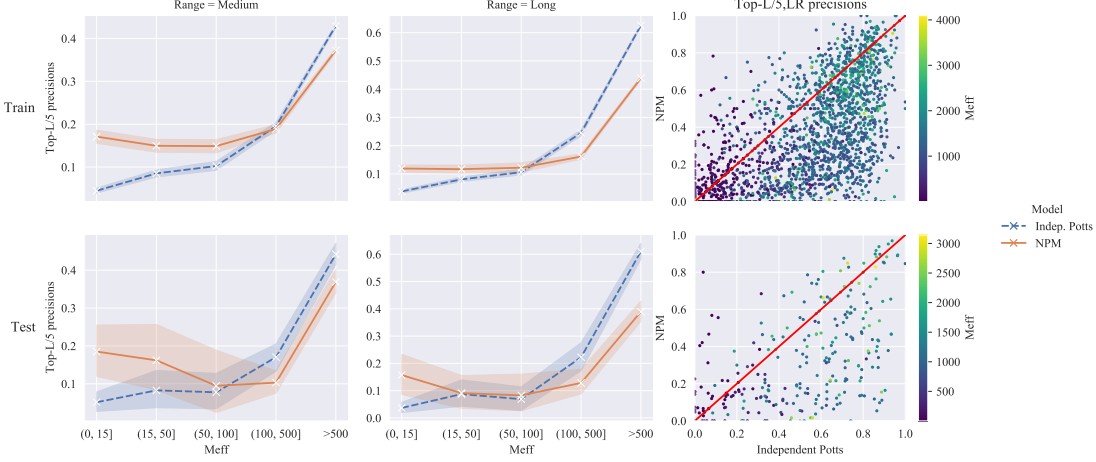

Figure 5: UniRef50: contact prediction precisions (higher is better) with 95% bootstrapped confidence intervals, on medium range (left), long range (middle), binned by MSA depth $M_{\text{eff}}$. Top row: sequences from the train set; bottom row: sequences from the test set. For shallow MSAs, average performance of NPM is higher than the independent Potts model. Right: scatter plot comparing long range precision from NPM vs independent Potts model, each point is a protein. More metrics are presented in Appendix D Fig. 10.

We now perform an evaluation in the more realistic setting of the UniRef50 dataset (Suzek et al., 2007). First we examine MSA depth across UniRef50 (Suzek et al., 2007). Appendix C.4 Fig. 7 finds that 19% of sequences in UniRef50 have MSAs with fewer than 10 sequences. (38% when a minimum query sequence coverage of 80% is specified).

We ask whether an amortization gain can be realized in two different settings: (i) for sequences the model has been trained on; (ii) for sequences in the test set. We partition the UniRef50 representative sequences into 90% train and 10% test sets, constructing an MSA for each of the sequences. During training, the model is given a sequence from the train set as input, and the NPM objective is minimized using a sample from the MSA of the input sequence. In each training epoch, we randomly subsample a different set of 30 sequences from the MSA to fit the NPM objective. We use ground-truth structures to evaluate the NPM couplings and independent Potts model couplings for contact precision. The dataset is further described in Appendix C.4; and details on the model and training are given in Appendix C.1.

The independent Potts model baseline is trained on the full MSA. This means that in setting (i) the NPM and independent Potts models have access to the same underlying MSAs during training. In setting (ii) the independent Potts model is afforded access to the full MSA; however the NPM has not been trained on this MSA and must perform some level of generalization to estimate the couplings.

Figure 5 shows a comparison between the NPM predictions and individual Potts models fit from the MSA. The Neural Potts Model is given only the query sequence as input. On top-L/5 long range precision, NPM has better precision than independent Potts models for 22.3% of train and 22.7% of test proteins. We visualize in Fig. 6 example proteins with low MSA-depth where NPM does better than the individual Potts model. For shallow MSAs, the average performance of NPM is higher than the Potts model, suggesting an inductive generalization gain.

To contextualize the results let us consider the setting where the amortized Neural Potts Model (i) matches the independent Potts model on training data: this means the NPM can predict good quality couplings from a single feedforward pass without access to the full MSA at inference time; (ii) surpasses the independent model on training data: the amortization helps the NPM to improve over independent Potts models, i.e. it realizes inductive generalization gain; (iii) matches the independent model on test sequences: indicates the model is able to synthesize a good Potts model for sequences not in its training data; (iv) surpasses the independent model on test sequences: the model actually improves over an independent Potts model even for sequences not in its training data. In combination these results indicate a non-trivial generalization happens when NPM is trained on UniRef50.

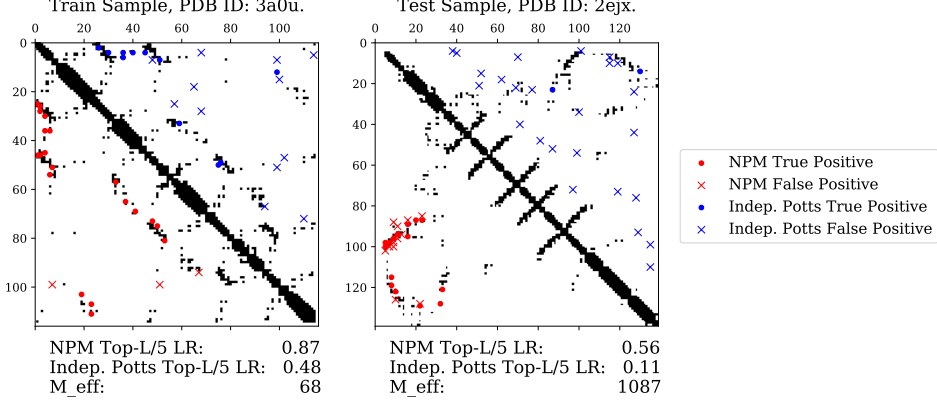

Figure 6: Examples where NPM outperforms the independent Potts model fit directly on the MSA. NPM top-L/5 LR contact prediction (lower diagonal, red) compared to the independent Potts model prediction (upper diagonal, blue). All ground truth contacts are indicated in black. True and false hits are indicated with dots and crosses, respectively.

## 5 RELATED WORK

Recently, protein language modeling has emerged as a promising direction for learning representations of protein sequences that are useful across a variety of tasks. Rives et al. (2019) and Rao et al. (2019) trained protein language models with the masked language modeling (MLM) objective originally proposed for natural langue processing by Devlin et al. (2019). Alley et al. (2019), Heinzinger et al. (2019), and Madani et al. (2020) trained models with autoregressive objectives. Transformer protein

language models trained with the MLM objective learn information about the underlying structure and function of proteins including long range contacts (Rives et al., 2019; Vig et al., 2020). This paper builds on the ideas in the protein language modeling literature, introducing the following new ideas: the first is supervision with an energy landscape (defined by a set of sequences) rather than objectives which are defined by a single sequence; the second is to use amortized optimization to fit a single model across many different energy landscapes with parameter sharing; the final is the consideration of the unsupervised contact prediction problem setting rather than the use of representations in a supervised pipeline.

Unsupervised structure learning is reviewed in the introduction. The main approach has been to learn a set of constraints from a family of related sequences by fitting a Potts model energy function to the sequences. Our work builds on this idea, but rather than fitting a Potts model to a single family of related sequences, proposes through amortized optimization to fit Potts models across many sequence families with parameter sharing in a deep neural network.

Supervised learning has produced breakthrough results for protein structure prediction (Xu, 2018; Senior et al., 2019; Yang et al., 2019). State-of-the-art methods use supervised learning with deep residual networks on co-evolutionary features derived from the unsupervised structure learning pipeline. While Xu et al. (2020) show that reasonable predictions can be made without co-evolutionary features, their work also shows that these features contribute significantly to the performance of state-of-the-art pipelines.

Prior work studying protein language models for contact prediction focuses on the supervised setting. Bepler & Berger (2019) studied pre-training an LSTM on protein sequences and fine-tuning on contact data. Rives et al. (2019) and Rao et al. (2019) studied supervised contact prediction from Transformer protein language models. Vig et al. (2020) found that contacts are represented in Transformer self-attention maps. Our work differs from prior work on structure prediction using protein language models by focusing on the unsupervised structure learning setting. It would be a logical extension of this work to integrate the Neural Potts model into the supervised pipeline.

## 6 DISCUSSION

This paper explores how a protein sequence model can be trained to produce a local energy landscape that is defined by a set of evolutionarily related sequences for each input. The training objective is cast as an amortized optimization problem. By learning to output the parameters for a Potts model energy function across many sequences, the model may learn to generalize across the sequences.

We also formally and empirically investigate the generalization capability of models trained through amortized optimization. We consider the setting of training independent Potts models on the MSA of each sequence, in comparison with training a single model using the amortized objective to predict Potts model parameters for many inputs. Empirically the amortized objective provides an inductive gain when few related sequences are available in the MSA for training the independent Potts model.

A number of direct extensions exist for future work, including further investigation of model architecture and parameterization of the energy function by the deep network, use of the amortized models in a supervised pipeline, and combining independent Potts models with amortized couplings. The hidden representations could also be investigated for structure prediction and other tasks using the approaches in the protein language modeling literature. The main contribution of this work is to directly incorporate information from a set of sequences related to the input in the learning objective. It would be interesting to investigate other possible approaches for incorporating this type of supervision into models that aim to learn underlying structure from sequence data.

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

# Appendix

## A   LEARNING THE NEURAL POTTS MODEL

---
**Algorithm 1** Learning the Neural Potts Model
---
Initialize the amortized model $W_\theta$
**while** unconverged **do**
    Sample $N$ sequences $\boldsymbol{x}_n \sim \mathcal{D}$ and $M$ MSA sequences $\tilde{\boldsymbol{x}}_n^m \sim \mathcal{M}(\boldsymbol{x}_n)$
    grad_update$(\theta, \nabla_\theta \hat{\mathcal{L}}_{\mathrm{NPM}}(\theta; \{\boldsymbol{x}_n\}, \{\tilde{\boldsymbol{x}}_n^m\})$       ▷ Optimize the regularized loss in Eq. (8)
**end while**
---

## B   MODEL ARCHITECTURE: MULTI-HEAD BILINEAR FORM FOR PAIRWISE COUPLINGS

In this Section, we describe the model architecture to compute a four-dimensional pairwise coupling tensor $J_\theta(\boldsymbol{x})$ from sequence embedding $g_\theta(\boldsymbol{x})$.

### B.1   MULTI-HEAD BILINEAR FORM

We write sequence length $L$ and amino acid vocabulary $K = 21$. The single site potentials $h \in \mathbb{R}^{L \times K}$, and the pairwise couplings are a four-dimensional tensor: $J \in \mathbb{R}^{L \times K \times L \times K}$ indexed as $J_{ij}(k,l)$.

We start with a sequence-level model to produce the embedding $e$ of the sequence (typically final hidden layer output): $e = g_\theta(\boldsymbol{x}) \in \mathbb{R}^{L \times d}$. The estimator for single-site potential $h_\theta(\boldsymbol{x})$ is a linear projection layer on the embedding; $h_\theta(\boldsymbol{x}) = g_\theta(\boldsymbol{x}) P^h$ with $P^h \in \mathbb{R}^{d \times K}$. Now we discuss how to parametrize the estimator $J_\theta(\boldsymbol{x}) \in \mathbb{R}^{L \times K \times L \times K}$.

**Multi-head bilinear form for direct prediction**   We introduce a *multi-head bilinear form* (mhbf) on the embedding $e$; i.e. for every pair $k, l$ of amino acids we have a bilinear form, parametrized with a learned interaction matrix $B^{kl} \in \mathbb{R}^{d \times d}$ connecting the hidden states at positions $e_i, e_j \in \mathbb{R}^{1 \times d}$. So we compute the $K^2$ bilinear forms for amino acid pairs $(k, l)$ between $L \times L$ position pairs $(i, j)$: $J_{ij}(k,l) = e_i B^{kl} e_j^\top$. We always use a low-rank decomposition $B^{kl} = U^{kl} V^{kl\top}$ with both $U^{kl}, V^{kl} \in \mathbb{R}^{d \times d'}$, so the bilinear form becomes the inner product in the lower-dimensional space with $d'$ the projection dimension: $(e_i U^{kl})(e_j V^{kl})^\top$. We can interpret this as an inner product of embeddings $i, j$ after linear projection to a space specific to amino acid pair $(k, l)$. This low-rank multi-head bilinear form is similar to the multi-head attention mechanism introduced in Vaswani et al. (2017), but without softmax normalization.

Notation-wise, our parameters $\theta$ include the parameters of the transformer that produces the embedding and the components of the interaction matrix $\{U^{kl}, V^{kl}\}$.

**Direct prediction: tied projection**   One way to reduce the number of parameters in the multi-head bilinear form, is for the low-rank decompositions of the $K^2$ heads $B^{kl}$ to share their decomposition per $k, l$. Instead we can *share/tie* the projection matrices between amino acids $k, l$: $U^{kl} = U^k$ and $V^{kl} = V^l$, such that head $B^{kl} = U^k V^{l\top}$. Note that the dot product in this case is after a linear projection specific to single-site amino acid $k$ and $l$ separately; $J_{ij}(k,l) = e_i B^{kl} e_j^\top = (e_i U^k)(e_j V^l)^\top$.

**Direct prediction: Symmetry**   We can or should parametrize the estimator of J to be symmetrical against interchanging both i,j and k,l: $J_{ij}(k,l) = J_{ji}(l,k)$, i.e. no difference between the order of considering interaction between AA $k$ at position $i$ with AA $l$ at position $j$. This does not mean symmetry of each interaction matrix! We ask that

$$J_{ij}(k,l) = e_i B^{kl} e_j^\top = e_j B^{lk} e_i^\top = e_i B^{lk\top} e_j$$

The second equality is the symmetry, the last equality by transposing the bilinear form. From $B^{kl} = B^{lk\top}$ it follows that

$$U^{kl}V^{kl\top} = V^{lk}U^{lk\top}$$
$$U^{kl} = V^{lk}$$

The last equality is the obvious choice. In the tied parametrization, this simply becomes $U^{kl} = U^k = V^k = V^{lk}$ such that $W_{kl} = U^k U^{l\top}$. Once again, note that the dot product now becomes $J_{ij}(k,l) = (e_i U^k)(e_j U^l)^\top$. We present a Tensor decomposition perspective on this multi-head bilinear form in Appendix B.2.

**Convolutional layers after multi-head bilinear form** As an extended model architecture, we consider having convolutional layers after the multi-head bilinear form (only used for the Pfam experiments). parametrized with a learned interaction matrix $B^{kl} \in \mathbb{R}^{d \times d}$ In this case, rather than having $K^2$ heads $B^{kl}$, we now have an arbitrary number of heads $F$ which will become the number of channels in the consecutive convolutional layers: $B^F = U^F V^{F\top}$. We add $1 \times 1$ convolutional layers having also $F$ channels, and finally $K^2$ output channels for the last convolutional layer. Weight tying and symmetry considerations of the mhbf do not apply in this model variation.

## B.2 TENSOR DECOMPOSITION VIEW ON MULTI-HEAD BILINEAR FORM

We can see the multi-head bilinear form as a tensor decomposition of $J$, for which we will use Einstein notation to indicate that any pair of indices appearing both in subscript and superscript are summed over their range. Let us write the tensor collecting the $U^{kl}$ matrices as $\mathcal{U} \in \mathbb{R}^{K \times K \times d \times d'}$; and index into $\mathcal{U}$ in the same notation as for $U$: $\mathcal{U}_{\alpha r}^{kl} = U_{\alpha r}^{kl}$. With $\alpha, \beta \in [1 \ldots d]$, $r \in [1 \ldots d']$ The same for $\mathcal{V}$. Now the J estimate in the full untied asymmetric case, written as tensor, becomes

$$J_{ij}^{kl} = e_i^\alpha \, \mathcal{U}_{\alpha r}^{kl} \, \mathcal{V}_\beta^{klr} \, e_j^\beta$$

or the symmetric ($\mathcal{U}_{\alpha r}^{kl} = \mathcal{V}_{\alpha r}^{lk}$) and tied ($\mathcal{U} \in \mathbb{R}^{K \times d \times d'}$) version:

$$J_{ij}^{kl} = e_i^\alpha \, \mathcal{U}_{\alpha r}^k \, \mathcal{U}_\beta^{lr} \, e_j^\beta$$

Note that the $\mathcal{U}, \mathcal{V}$ are shared across proteins, while the embeddings $e = g_\theta(\boldsymbol{\sigma})$ are specific per protein, based on a high-capacity sequence level model.

## C EXPERIMENT DETAILS

### C.1 TRAINING DETAILS

We summarize the precise model architecture and optimization settings in Table 2. During each NPM training step, for a given input $x$, M sequences $\tilde{x}^m$ are randomly sampled (M=100 or 30, see Table 2), for the pseudo-likelihood loss evaluation in Eq. (8). Each sequence is selected with probability according to its sequence weight $w^m$. One can think of these $M$ sampled sequences as similar to a minibatch. Note that to compute the independent Potts model baseline, the Potts model is computed without any downsampling of the MSA. Additionally, in the Pfam experiments the loss term for family $n$ in Eq. (8) is upweighted with a factor $\sqrt{M_{\text{eff}}(n)}$, which places more weight on the well-formed, deep MSAs and discounts the shallower MSAs. In both the Pfam and UniRef experiments, we enforce a max sequence length of 512 via random contiguous crops of positions.

### C.2 VALIDATION DETAILS

To compute precisions, we convert the pairwise couplings $J \in \mathbb{R}^{L \times K \times L \times K}$ to an $L \times L$ pairwise coupling score by (1) zeroing all positions in $J$ corresponding to gap characters, (2) computing the magnitude via Frobenius norm over the $K \times K$ matrix $J_{ij}$ for every pair of positions $i, j$, and (3) applying the Average Product Correction (Dunn et al., 2008). True contacts are defined as pairwise distances less than or equal to 8 Angstroms. Precision is calculated as the true positive fraction of the top $L$, $\lfloor L/2 \rfloor$, or $\lfloor L/5 \rfloor$ predicted contacts. Additional to precision, the Area Under the

| | | Pfam | UniRef50 |
|---|---|---|---|
| Model details | Parameters | 111M (frozen: 26M trained) | 519M |
| | Number of layers | 12 | 32 |
| | Embed dim | 768 | 1024 |
| | Attention heads | 12 | 16 |
| | mhbf $U, V$ projection dimension | 128 | 512 |
| | mhbf number of heads | 128 | $K^2 = 441$ |
| | Conv after mhbf | $1 \times 1$ (3 layers) | No |
| | mhbf tied | N/A | No |
| MLM pre-training (UniRef50) | # of epochs | 21 | 50 |
| | # of gradient updates | 175k | 449k |
| | Learning rate | 4e-4 | 3e-5 |
| NPM training | Number of updates | 10k | 135k |
| | Adam learning rate | 1e-4 | 3e-5 |
| | $\rho(W)$ multipliers $\lambda_J, \lambda_h$ | 1e-3, 1e-2 | 1e-3, 1e-2 |
| | M (samples per MSA) | 100 | 30 |
| | Batch size (# seqs) | 64 | 2048 |
| | Max sequence length | 512 | 512 |
| | Base model frozen | Yes | No |

Table 2: Hyperparameters

Precision-Recall Curve (AUC) is computed, summing over thresholds stepwise per $L/10$ increment up to $L$. Precision and AUC metrics are computed at sequence separations $s$ of short ($6 \leq s < 12$), medium ($12 \leq s < 24$), and long ($24 \leq s$) ranges.

For the independent Potts model baselines in all experiments, we use CCMpred (Seemayer et al., 2014), a GPU implementation of pseudolikelihood maximization (Balakrishnan et al., 2011). The coupling matrix $J$ from the independent Potts model is processed in the same way following steps (1-3) described above.

### C.3 PFAM TRAINING DATA AND SETUP

**Data Selection.** We use the Pfam database (Finn et al., 2016) version 28.0. All MSAs in the HTH (n=217), P-loop NTPase (n=198), NADP Rossmann (n=181), and AB hydrolase (n=67) clans were parsed from the multiple alignment file Pfam-A.full. We apply two preprocessing steps to all MSAs. First, for speed, we only load up to a maximum of 100k sequences from each MSA. Next, we apply HHfilter, from the HHSuite3 (Steinegger et al., 2019) toolset, with all default settings, to each MSA. We find that filtering improves contact prediction accuracy of the independent Potts model baseline.

**Dataset splits.** We perform the experiment using a five-fold cross-evaluation scheme, in which we partition the clan's families into five equally-sized buckets. As in standard cross-validation, each bucket will eventually serve as an evaluation set. However, we do not remove the evaluation bucket, but artificially reduce the number of sequences in the MSAs in the evaluation bucket to a small fixed MSA depth (=purging the MSA). All Pfam experiments are repeated 5 times, each with a different selection for the reduced bucket. In our figures, we plot average results, with confidence interval bounds defined by the standard deviation across the five-fold cross-evaluation.

**During NPM training**, we iterate over the the set of MSAs in the four buckets that have not been reduced, as well as the reduced bucket. At a given training step, we randomly select a sequence $x$ within an MSA for use as input to NPM. This selection is likely to return a sequence with inserted gap characters. We drop these gap characters and their corresponding columns in the MSA. Then we randomly subsample 100 sequences $\tilde{x}$ from the MSA to fit the NPM objective. The procedure is described in more detail in Appendix C.1.

**Evaluation** During evaluation, we assess the NPM and the independent Potts model via a contact prediction task (described in previous subsection), on the families in the evaluation bucket. For

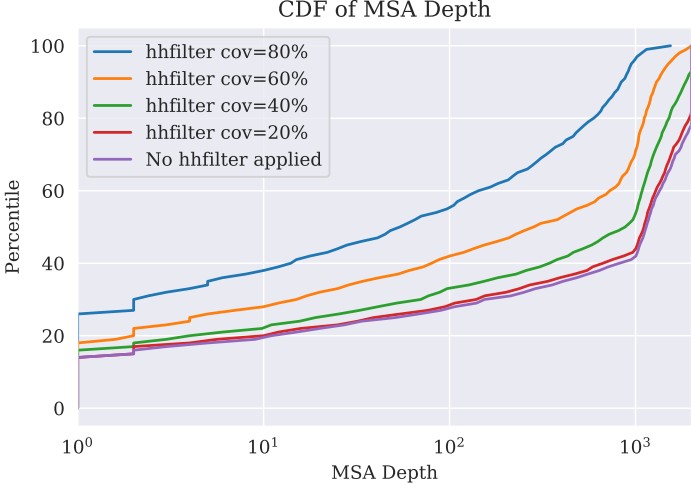

Figure 7: A random sample of 1000 MSAs for 1000 sequences in UniRef50 were analyzed. The graph shows cumulative density plots with MSA depth in log-scale on the x axis, for different query sequence coverage requirements (-cov [20, 80]) specified by a call to HHfilter, applied to each MSA. The fraction of MSAs with depth ≤ 10 is 19% (38% when a coverage of 80% is specified), while the fraction of MSAs with depth ≤ 100 is 30% (55% when a coverage of 80% is specified) .

each family, a single structure is selected as target, using the pdbmap included in Pfam. NPM's contact predictions are made using only the sequence belonging to the target structure. To compute the independent Potts model for a given family in the evaluation bucket, the depth-reduced MSA is aligned to the sequence from the target structure, and the Potts model is computed without further downsampling.

As an additional baseline, we predict contacts for validation sequences using the Potts model of the "Nearest Neighbor" family in the train set. For a given validation sequence, we calculate "nearness" to all train families via calls to HHalign given the sequence and the train family's Pfam seed alignment as input. We select the family with the highest HHalign probability score as the nearest neighbor. The nearest neighbor prediction is generated as follows: (1) the validation sequence is aligned to the selected train family's MSA; (2) an independent Potts model is fit to the selected train family's MSA, yielding a predicted contact map for the train family; (3) the rows and columns of the predicted contact map that align to the validation sequence are extracted to construct a prediction for the validation sequence.

### C.4 UNIREF50 TRAINING DATA AND SETUP

For the experiments in Section 4.2, we retrieve the UniRef-50 (Suzek et al., 2007) database dated 2018-03. The UniRef50 clusters are partitioned randomly in 90% train and 10% test sets. For all sequences, we construct MSAs using HHblits (Steinegger et al., 2019) against the UniClust30_2017_10 database. HHblits is run using the default settings, for 3 iterations with an e-value of 0.001.

It is important to note that given this MSA generation procedure, validation sequences can be included in MSAs of train sequences. However, we are guaranteed that validation sequences are not trained on as input to NPM.

**Evaluation of contact precision**

We use contact precision as a proxy to measure unsupervised structure learning in the underlying Potts model. To define a set of structures for evaluation, we collect structures from the PDB, and assign them to either the training sequences or test sequences. This allows us to separately examine performance of NPM on sequences from its training set, and sequences from its test set. Note, that

the structures are used only to evaluate unsupervised contact prediction performance of the model; the model is never trained on structures.

We query the Protein Data Bank (PDB) to obtain a list of all protein structures with a resolution less than 2.5 Å, a length greater than 40 residues, and a submission date before May 1, 2020. We search each pdb entry for hits against the sequences in the training and test sets for NPM respectively. If the PDB entry retrieves hits only to training sequences we assign it to the training-sequences group. If the PDB entry retrieves hits only to test sequences we assign it to the test-sequences group. Any PDB entry which hits both training and test sequences or neither, is discarded. To perform the search we use the MMSeqs2 software suite (Steinegger & Söding, 2018) using the default settings with 50% sequence identity at 80% target coverage. We then cluster each of the two groups of PDB entries at 50% sequence similarity, resulting in a dataset of 11040 structures assigned to train sequences and 211 structures assigned to validation sequences. MSA construction for the PDB entries precisely follows the procedure for UniRef50 (first paragraph); the method for contact prediction from the model couplings (for NPM or the independent Potts model) is described in Appendix C.2.

# D    ADDITIONAL EXPERIMENTS

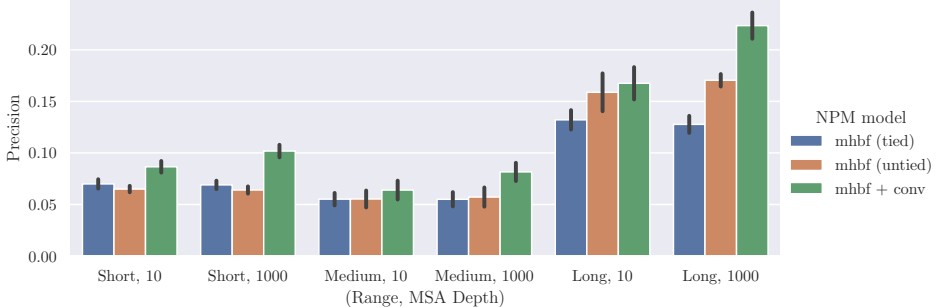

Figure 8: Comparison of the main NPM model architecture choices, evaluated using the Pfam experimental setup. We show precisions at fixed top-L threshold, while on the x-axis we vary sequence separation range and two levels of MSA depth reduction (10 and 1000). Standard deviations over the five-fold cross-evaluation are shown. For the direct multi-head bilinear form (mhbf) prediction (tied or untied), we found an improvement from using $U, V$ projection dimension 512 rather than 128. Other hyper-parameters follow Table 2.

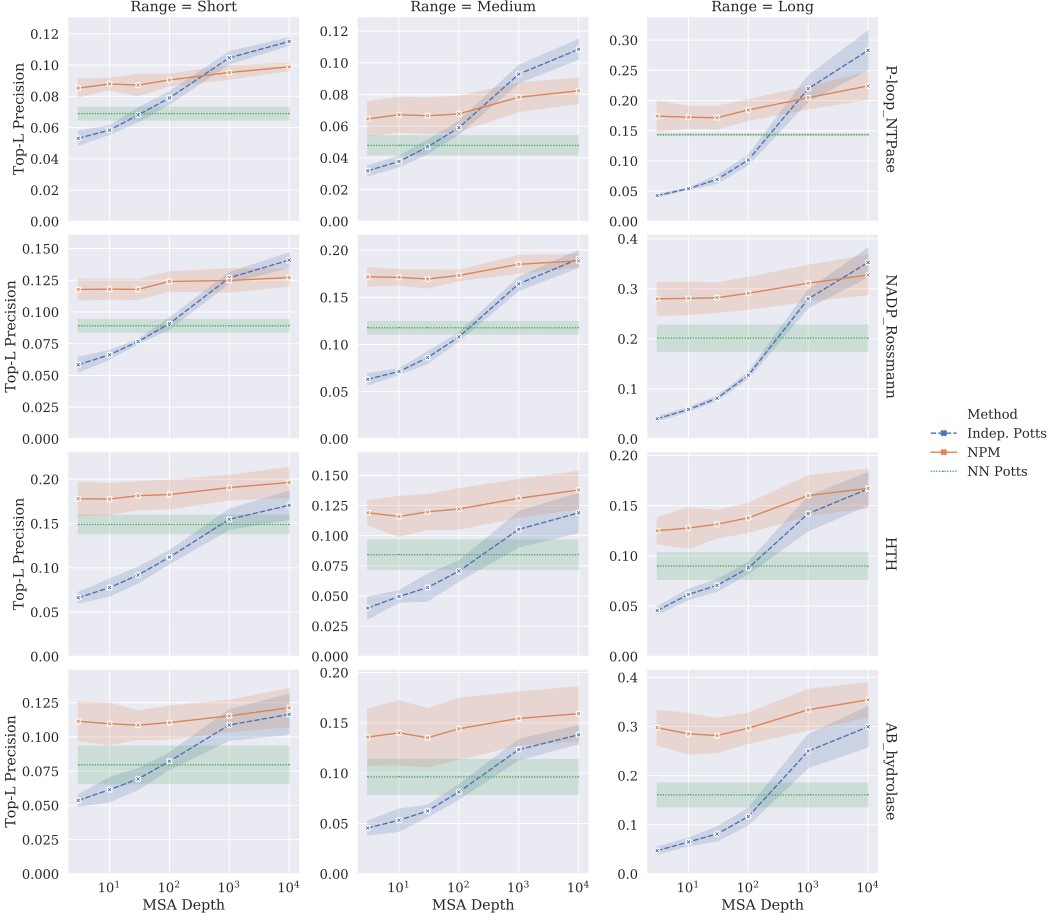

Figure 9: Contact prediction precision on Pfam families at different levels of depth reduction. Three approaches are compared: (1) the Neural Potts Model, (2) independent Potts Model, (3) Nearest Neighbor Potts. On shallow MSAs, NPM outperforms both independent Potts and Nearest Neighbor Potts for all four of the clans. On deep MSAs NPM matches the independent Potts baseline for 3 of the 4 clans. Clans are noted on the right hand side.

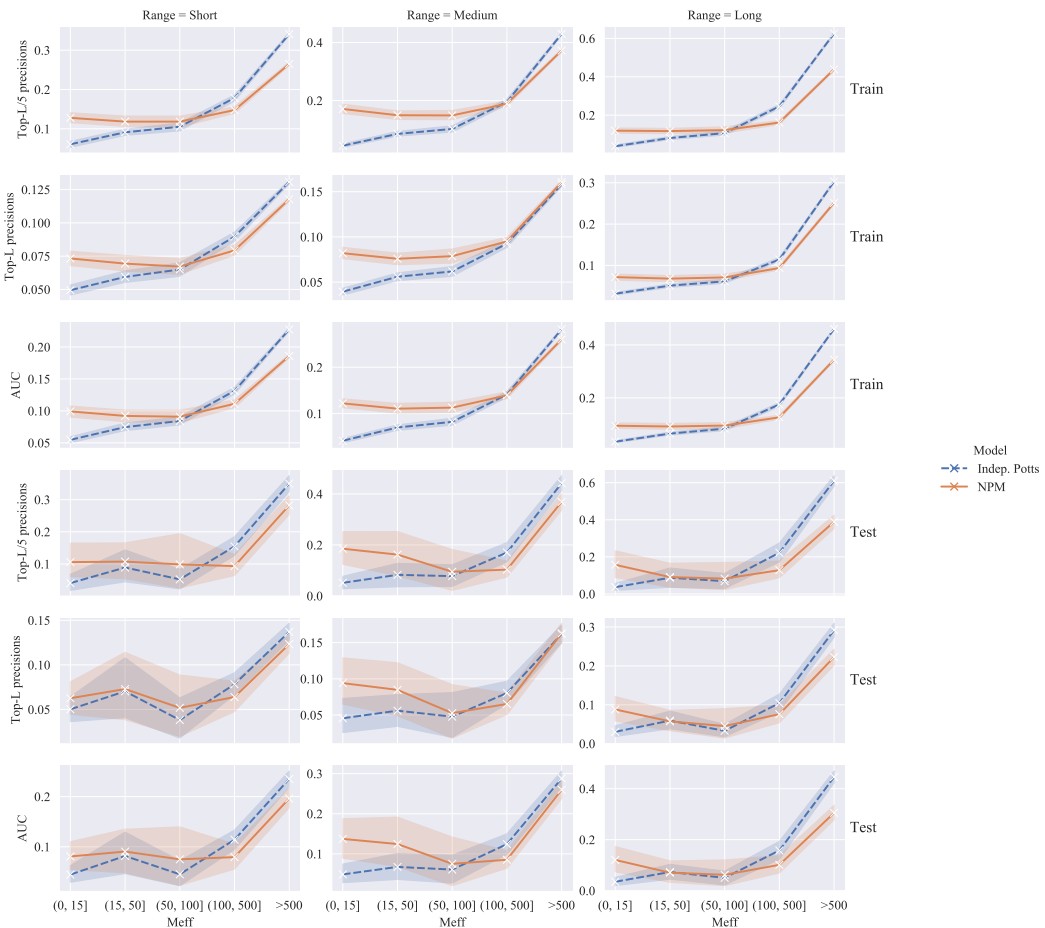

Figure 10: Additional metrics for the UniRef50 results. We show the sequence separation range as columns. On each row, we vary over cutoff thresholds L, L/5 and AUC defined in Appendix C.2. The top 3 rows are for train sequences, bottom 3 rows for test sequences.

