# OpenReview forum: "Neural Potts Model"
_ICLR.cc/2021/Conference — Reject_

### Official Review · AnonReviewer3 · 2020-10-15
**Interesting idea. Limited experimental evidence.**

**Rating:** 6
**Confidence:** 4

**Review:**

This paper aims to improve low-depth MSAs, when a protein of interest only has a small number of known evolutionarily related sequences. This is a well motivated problem. MSAs are commonly used for a variety of purposes. Methods to enhance low-depth MSAs can be very useful. In particular, this paper focuses on using MSAs for contact prediction as a down-stream task. I'm not sure if contact prediction is the best use case for this, but it's a well-studied task for proof of concept.

The weight-sharing Neural Potts Model approach is novel (as far as I know) and is a well-motivated idea. There could be reasons to believe that weight sharing could improve Potts Model compared to directly fitting Potts Model to individual MSAs. Potts Model is very commonly used and it makes sense to base their method on Potts Model.

Personally, I think this approach is interesting, but the paper needs further experimental validation for acceptance. There are two main limitations in the experiments:

   1. The paper claims that there is a gain from amortization for low-depth MSAs. This claim is only partially supported for very shallow MSAs. According to the experiments in the paper, there is only a gain for very shallow MSAs (fewer than 15 sequences) for long-range contact prediction, or for MSAs with fewer than 50 sequences for medium-range contact prediction. I'm not sure whether it's a common situation to have only 15 sequences in an MSA. It would be helpful if the authors could provide example use cases where this applies.

   2.  The UniRef50 clusters are based on 50% sequence identity. In the paper, the train-test split is based on the clusters. There could still be significant information leakage if some train and test sequences share 40% similarity. I would like to see the authors further examine this and show that this is not the case.

Suggestions for improving the paper:

  1.  Include more than one PFAM clan level study. The P-loop NTPase example is well taken. It seems like the results here are quite different from the UniRef50 results (NPM has advantage for up to MSA depth 100s vs 15). This could be explained by the families in the same clan are a lot more similar than in UniRef50 in general. It would be helpful to understand why the NPM approach worked particularly well on NTPase, and whether it works similarly well on other PFAM clans. It seems like instead of aiming to use the method on arbitrary sequences, the method could be more valuable in cases where we have MSAs for other related families in the same clan.

  2.  Split train-test differently in UniRef50 to make sure similar families don't end up in both train and test.

  3. Instead of only evaluating the top L/5 contact prediction task, try other thresholds. The achieved contact prediction precision is around 10-20%. That seems rather low for both NPM and direct Potts, so hard to say if the gain is meaningful.

  4. (Perhaps out of scope for this paper.) Could this method be applied to generate Potts Model parameters to be fed into other structure prediction models (e.g. AlphaFold). For example, it would be interesting to see that the NPM Potts Model can improve AlphaFold contact prediction results.

  5. References for important proteins with shallow MSAs.

Clarification questions:

1. How do you compute the MSA? Is it from HHblits or some other standard package?
2. In the PFAM experiment, how do you vary the MSA Depth? Is it random subsampling?

--------------------------------------------------------------------------------------------------------------
Update:
Updated score from 5 to 6. The author response clarified some key questions and the updated paper incorporated some of the feedback.

---

> ### Author Response · Authors · 2020-11-21
> **Individual response to R3**
>
> We thank the reviewer for their detailed feedback and interest in the approach.
>
> > the paper needs further experimental validation for acceptance
>
>
> Thank you for all the great suggestions. We will add additional experiments (detailed below) in response to your comments.
>
> > 1. Shallow MSAs
>
> This is a problem setting known to be important in the literature. See e.g. the review by Tetchner et al. cited in the general comment (ii) for background on limitations of existing methods and importance of proteins with low MSA depth. Additionally we reference Figure 6 for the prevalence of low-depth MSAs (19-38% of sequences in UniRef50 with MSA depth <10). We find that independent Potts models perform poorly in this regime (as is already known), and that the NPM shows an advantage, confirming the core theoretical contribution (inductive generalization gain) can be realized in this setting.
>
> > 2. UniRef50: information leakage
>
> Please refer to the general comment (iii). Please note that the model is trained only on sequences, without structure supervision. Because it is a purely unsupervised method there is no labeled data that could leak between the train and test splits. Also note that the independent baselines (at test time) and neural model (at train time) have access to the same underlying sequence database so they are on equal footing when they are evaluated. We don’t claim to measure generalization between structural families; the goal of the experiment is simply to show the model can synthesize reasonable couplings for sequences outside its training set.
>
> > PFAM vs UniRef50:
>
> This is a great observation. Indeed PFAM is a much simpler task: the model (despite being smaller than the UniRef50 model) has enough capacity to well amortize all Potts models. In contrast UniRef50 spans a much wider diversity of families, so even the larger model does not have enough capacity to perfectly fit the UniRef families. More parameters and better architectures can likely improve this.
>
> > More than one pfam clan
>
> Thanks for this suggestion, we will add additional experiments on HTH and AB_hydrolase clans.
>
> > UniRef50 Split
>
> Please refer to the general comment (iii), and comment above.
>
> > only evaluating the top L/5 contact prediction task, try other thresholds
>
> In the revision we will provide both the top-L and L/5 metrics, as well as an AUC computed over different thresholds in the Appendix.
>
> > Potts Model parameters to be fed into other structure prediction models
>
> This is a really interesting suggestion and we agree that the integration of the Neural Potts Model in supervised pipelines is a great opportunity for future work. Our preliminary experiments here were inconclusive.
>
> > References for important proteins with shallow MSAs.
>
> See general comment (ii).
>
> > How do you compute the MSA?
>
> For all sequences, we construct MSAs using HHblits (Steinegger et al., 2019) against the UniClust30_2017_10 database. HHblits was performed for 3 iterations, with an e-value of 0.001. (See Appendix C.4)
>
> > PFAM: how do you vary the MSA Depth?
>
> Thanks for the question, we will add this detail in Appendix C.3. We keep the top-M sequences in the MSA, after applying HHfilter to filter too redundant sequences and increase diversity (See Appendix C.3). The MSA sequences are ordered by e-value, so we keep the most maching sequences.

---

### Official Review · AnonReviewer4 · 2020-10-28
**An amortized optimization framework to learn a mapping between protein sequence and the parameters of a Potts model describing the local energy landscape of the input sequence. Approach is well motivated and addresses limitations of existing approaches when few sequences are available in the neighborhood of the target sequence. Paper is well written and easy to read. Experimental results demonstrate that the approach works well in the regime for which it was introduced (low density regime)**

**Rating:** 7
**Confidence:** 4

**Review:**

Overall comments
- The paper introduces an amortized optimization framework to learn a mapping between an input sequence and the coefficients of the corresponding Potts model -- a standard method used in computational biology to model protein sequences.
- The idea is particularly compelling in the regime where there exists a limited number of sequences (in the training data) that are similar to the target sequence. That would typically result in a low quality multiple sequence alignment (MSA), and in turn a poor Potts model.
- The idea is sound from a theoretical standpoint. The newly introduced framework is well motivated: authors did a good job at explaining the background for the problem and providing an intuition for why the proposed amortized optimization framework would improve over the current approach (i.e. independently training models on each MSA).
- The paper is very well written and easy to read -- the language is precise, everything is defined very clearly.
- Experiments show good results in the regime that matters most for the introduced Neural Potts model (NPM), i.e., low density of sequences around the query.
- The approach would be even more compelling if the NPM would not underperform the baseline in the regime where Meff (the “effective number” of sequences in the MSA) is larger.
- Bridging that gap would be very compelling from a practical standpoint: large scale studies typically involve modeling thousands of distinct proteins, and thus fitting thousands of distinct models independently. Intuitively, there is shared information between these models, which is lost when models are trained independently but could be captured via a process akin to the one introduced in this paper.

Detailed comments and questions

Section 1
- Would suggest to add one sentence describing why it matters to solve the low Meff regime to further motivate your approach (e.g., which scientific questions would we be able to better answer?).

Section 2
- I would add a paragraph on Transformers models for protein sequence embedding (since it is a core component of the NPM used in experiments), and cite a few of the key works in this area, for example:
- Rives et al., Biological Structure and Function Emerge from Scaling Unsupervised Learning to 250 Million Protein Sequences (https://www.biorxiv.org/content/10.1101/622803v1)
- Madani et al., ProGen: Language Modeling for Protein Generation (https://www.biorxiv.org/content/10.1101/2020.03.07.982272v2)
- Vig et al., Bertology meets biology: Interpreting attention in protein language models (https://arxiv.org/abs/2006.15222)
- Rao et al., Evaluating Protein Transfer Learning with TAPE (https://arxiv.org/abs/1906.08230)

Section 3
- Is there a particular reason for which you are using a vanilla Transformer architecture? Other Transformer architectures (e.g., BERT-like architecture) may be able to learn better sequence embeddings and in turn further close the gap with the independent Potts model in the large Meff regimes (see the above papers for example).

Section 4
- How do you interpret the gap with the baseline Potts model approach in the medium-large Meff setting?
- Is there a way to detect (based on training) the situations where the NPM will likely underperform the independent Potts model baseline?
- Computationally, how expensive is it to train one NPM Vs one Potts model? For example, if I have 1000 proteins to model in the regime where both perform the same (Meff ~100), is it faster for me to train one NPM model or 1000 Potts models? I suspect the 1000 Potts models are still cheaper to train, but if not that could be one strength of your approach to mention.
- What is the distribution of Meff over the data (section 4.2)? Based on the rightmost plots it seems that the majority of MSAs are in the “>500” regimes where NPM underperforms independent Potts models (asking that question with the understanding that you are more interested by addressing the low Meff setting here).
- Do you have an intuition for why the NPM appears to be doing better for medium range interaction here Vs long range in experiment 4.1?
- How do you explain the U-shape of the top-L/5 precision curves for NPM (on heldout data)? I would have expected that increasingly larger Meff would be associated with monotonically increasing performance (as is the case for the Independent Potts model curves in blue).

Appendix C1
- Why the upweight by a factor sqrt(Meff(n))?

Appendix C3
- Did you look at the performance gap on HTH? Since baseline Potts models yield poor long-range contact on that dataset, perhaps NPM would have helped address that issue, given increased ability to model long range dependencies as per 4.1. It was not clear whether the low performance is due to an intrinsic limitation of the Potts model or potentially due to a low density around each query point in HTH (in which case NPM may help).

---

> ### Author Response · Authors · 2020-11-21
> **Individual response to R4**
>
> We thank the reviewer for the detailed feedback. Let us provide answers on the questions in bullet point form:
> * *Motivation for the shallow-MSA regime*:
> See the general comment (ii). We will indeed add some discussion in introduction along the lines discussed there.
> * *Paragraph about protein language models*:
> We agree with the reviewer and are adding some discussion in the main text.
> * *Vanilla Transformer*:
> We will clarify. We are using a bidirectional transformer as in BERT. We use a BERT/MLM-pretrained transformer (following Rives et al. 2019) to initialize the NPM.
> * *Gap on deeper MSAs*: we believe that the gap on deeper MSAs is mainly due to the limited capacity/parameter count of the single model which has to capture a large variety of Potts models. For comparison, a single independent Potts model for a sequence of length 500 has approx 110M parameters (500 squared by 21 squared), though heavily overparametrized and regularized.
> * *Detect when NPM will underperform independent Potts model*: This is an interesting question for future work. The most obvious candidate would be the primary train objective (pseudo-likelihood loss on the MSA); however this quantity is heavily dependent on regularization (which varies between the independent Potts model and NPM), and wasn’t useful in our preliminary experiments.
> * *How expensive is it to train one NPM Vs 1000 Potts models*:
> This is a good question - computational cost will depend on the architectural details (depth and width mostly) of NPM. Usually NPM will converge in fewer iterations through the data. However, there is a clear savings in *disk space* to actually store the parameters of one NPM versus 10k Potts models (typical for supervised training); since each independent Potts model is on the order of 1-100M parameters, storing 10k Potts models consumes several TB of disk space (vs a few GB for NPM).
> * *Distribution of Meff*: (see also general comment (ii) ). The distribution of msa depth (M, upper bounds Meff) in UniRef50 is in Appendix, Figure 6, and shows that 19-38% of the UniRef50 sequences have <10 sequences in their MSA, and 30-55% of the UniRef50 sequences have <100 sequences in their MSA.
> * *Upweighting with sqrt(Meff)*: this reweighting causes the optimization to pay more attention to well-formed, deep MSAs and discount the shallower MSAs, which we found to be helpful. It can be seen as a middle ground between the vanilla formulation with (1/Meff), where each MSA contributes equally independent of its effective depth, and dropping the (1/Meff) completely which would make each MSA contribute proportional to its effective depth.
> * *performance gap on HTH*:
> Thanks for the suggestion. We will add additional experiments on HTH and AB_hydrolase clans.

---

### Official Review · AnonReviewer2 · 2020-10-28
**Interesting problem, however, more validations required to strengthen the main claim.**

**Rating:** 6
**Confidence:** 4

**Review:**

The paper proposes a new object called Neural Potts Model (NPM) to train a Transformer to learn the local energy landscape of protein sequences. The problem of modeling energy landscapes using the power of techniques in natural language processing (NLP) is a timely and interesting problem. However, there are some concerns that limit the strength and the main claim of the paper that needs to be addressed.

Is the NPM objective in equation (7) derived from some probabilistic model? While the intuition behind the objective can sound plausible from the discussion that follows, I am still curious what problem formulation is this objective solving? Is it a proper likelihood term?

Since the paper is advocating the use of NPM objectives, is it possible to plot the value of the proposed NPM objective in the Transformers as a function of the Top-L precision? This will strengthen the claim in proving all the gains are actually coming from the objective and not other possible factors in training the model.

The paper can be improved by more effort in punctuation and proper labeling of figures. There are many sentences and phrases that need a comma for better readability. There are extra parentheses when referencing Figures. What are the axes showing in Figure 2? What are the highlighted shades indicating Figure 3 (std or sem)? Is it possible to have error bars for Figure 4? In abstract, MSA is not defined.

Also given that there is sufficient space left in the paper, some material including the Algorithm box can be moved to the main text. Also, the author can consider to use the space to expand on the significance and importance of the NPM objective.

** after rebuttal: thanks for addressing the comments. I have revised my score based on the discussion.

---

> ### Author Response · Authors · 2020-11-21
> **Individual response to R2**
>
> We thank the reviewer for the feedback.
> > Is the NPM objective derived from some probabilistic model?
>
> In brief, no, we derived the Neural Potts Model as amortized optimization (Section 2.1) of a large collection of Potts models optimized with pseudo-likelihood maximization, see Section 2.
>
>
> > plot the value of the proposed NPM objective in the Transformers as a function of the Top-L precision
>
> This is a great suggestion. We are adding a plot of the NPM loss value against contact top-L precision, computed on the reduced-MSA bucket during PFAM training. It shows a monotonously decreasing NPM loss and increasing contact precision over the course of NPM training. This is in line with the common usage of the (non-amortized) independent Potts model pseudo-likelihood maximization as proxy loss for downstream contact prediction [1,2].
> > punctuation and labeling of figures
>
> Thanks for the detailed read and suggestions. Figure 2 shows a 1D cartoon of the loss landscapes (train and generalization) in function of the Potts model parameters. Figure 3: “(we  show) averages and standard deviations across (cross-evaluation) rounds”.
>
> * [1] Balakrishnan et al. (2011). Learning generative models for protein fold families.
> * [2] Ekeberg et al. (2013). Improved contact prediction in proteins: Using pseudolikelihoods to infer Potts models.

---

### Official Review · AnonReviewer5 · 2020-10-28
**NPM is a promising idea but the paper needs more work**

**Rating:** 6
**Confidence:** 3

**Review:**

###########################################################################################################


Summary of Paper
The motivation for the paper is a bit unclear. In the introduction, the authors begin by claiming to extend self-supervision "to information from a set of evolutionarily related sequences". However, it does not appear that the model is at all used for pretraining / representation learning as would be expected of a self-supervision method. No further connections to self-supervision are made in the rest of the paper. Based on the conclusion it seems that self-supervision is a future direction. If so, I would suggest rewriting the introduction to more clearly state the motivation of this paper.

The rest of the paper is more clear. The authors tackle the problem of predicting protein contacts. Standard unsupervised approaches fit a Potts model (often with pseudolikelihood) and then use the parameters of the fit model to make predictions about contacts. Such approaches require a MSA of reasonable depth and the amortized optimization approach suggested by the authors has the potential to work better than standard unsupervised approaches in the small-MSA regime by sharing information from all the protein families that the model is fit on.

The authors propose a meta-learning approach wherein a neural network (here a transformer) takes as input a single sequence and outputs the parameters of a Potts model. The model is trained with a pseudolikelihood loss across all sequences within individual families. This is a new and clever idea.

The main experimental result is that the NPM outperforms standard Potts models for shallow MSAs. This demonstrates that there apears to be some utility to this approach. However, the precision is still very poor and the authors make no claim that the enhanced accuracy is of biological utility (i.e. can be used to fold the protein). There are other tools that perform contact prediction from a single sequence with higher accuracy than the NPM. Furthermore NPM performs worse than standard Potts models for medium and large MSAs. This seems to be a significant drawback of the method.

#############################################################################################


Questions / Suggested Experiments:

PFAM EXPERIMENT:

Can the authors please explain why the PFAM experiments are evaluated on top-L precision whereas the remaining experiments are top-L/5 precision? The lack of explanation in the paper suggests the metrics may be cherry-picked to best show the performance of the model.

Since all the proteins in the family share structural similarity, it is not clear that NPM has learned to transfer any information. After looking at the structures in PyMOL it seems that significant substructure is shared between the different families. Here is a resonable baseline/experiment to add: Align the query sequence to the higher-depth MSAs and then using the MSA that best fits the query sequence, predict contacts based on the individual Potts model trained on that family. This would elucidate whether or not NPM is simply treating the query sequence as if it were from one of the higher-depth MSAs.

UNIREF EXPERIMENT:

The details of this experiment could use more explanation. I did not see any explanation of the purpose of the training and heldout sets, which in this setting are not obvious. My guess is that the authors are trying to demonstrate that the NPM can generalize zero-shot to new families? If so, please mention it in the text. Random splits are problematic for demonstrating zero-shot generalization since there could be very similar families in the train and heldout sets (e.g. same PFAM clan or structural class). As written, the paper barely discusses the purpose of the heldout set but if it is meant to convey some test of generalization the authors should more carefully construct their heldout test set accordingly.

Why is there no plot for short-range predictions, as there was for the PFAM experiment?

There is an unclear statement: "During training, we iterate over all sequences and their MSAs on every epoch, and subsample to M=30 sequences per MSA." Is this just for NPM or also for the Potts model? I would be very concerned if this is also what is done for the Potts models as it would significantly harm their performance. The authors need to clarify this in the text of the paper.

The authors do not discuss other existing methods that currently exists for shallow MSAs. For example, standard semi-supervised methods are fine-tuned on contact prediction and thus, work with single-sequence inputs. I believe these methods may perform as well or better than NPM. The authors should discuss these methods and include comparisons.


Model Architecture
Please do the following:
1) In Appendix B.1 you describe a number of "tricks" to reduce the number of parameters. These include (1) a low-rank decomposition of the bilinear form, (2) weight tying by amino acid for the decompositions (3) Convolutional layers. Please provide experiments showing the utility (or lack thereof) of each of these. This will lift the results from "here is what happens if you use these tricks" to "here is data suggesting that these choices we made are better than the naive choices". Thus, add more impact to the paper.

2) No explanation is given for why the architectures are so different for PFAM and UniRef (e.g. convolutional layers for PFAM only). Please provide an explanation. A priori, I see no reason to use different architectures for these two different domains. It seems to only complicate the paper and methods. If these choices really are important, then this seems to suggest overfitting to particular tasks rather than having a general solution.


Appendix Training Details:
I personally found the written descriptions here to be too vague to understand the exact details. This made it harder to evaluate the paper. Can you please clarify:
1) "we randomly subsample the MSA down to 100 sequences as NPM target sequences...". Based on equation (7) this means the pseudo likelihood is evaluate on 100 random sequences from the MSA. This seems to directly contradict the experiment shown in Figure 3 where the number of sequences in the MSA is varied up to 1000. Please explain what is happening here?


###########################################################################################

Explanation of Score

This paper was difficult to score because I find the idea / approach to be very creative and different from standard techniques of borrowing the latest NLP pertaining task and applying it to proteins. I applaud the authors for taking a unique approach.

The main drawback is that the authors exclusively evaluate the model on contact prediction and do not demonstrate convincing performance. While the model sometimes outperforms standard approaches with shallow MSAs the performance is still quite bad and no comparisons to other methods are shown.

Thus, overall I am impressed with the direction of the paper but think it needs more work.


Updated score from 5 -> 6 after clarifying feedback from the authors.

---

> ### Author Response · Authors · 2020-11-21
> **Individual response to R5**
>
> We thank the reviewer for the detailed feedback.
> > Summary
>
> We will clear up the framing wrt self-supervised learning. We are glad to see the reviewer appreciates the novelty of the core approach.
> Regarding the utility and significance of the results, please refer to the general comment (ii). In summary: we reference Figure 6 for the prevalence of low-depth MSAs (19-38% of sequences in UniRef50 with <10 sequences in their MSA), highlighting that independent Potts models perform poorly in this regime, and that the core theoretical contribution (inductive generalization gain) is experimentally confirmed. We outline avenues to improve the practical utility e.g. model improvements and scaling, integration in supervised pipelines, and combining NPM with independent Potts models.
>
> **PFAM**
>
> > top-L vs L/5
>
>
> Thanks for calling our attention to this - it was an oversight. In the revision we will provide both metrics as well as an AUC computed over different thresholds in the Appendix.
>
> > new baseline experiment:
>
> Thank you for a great suggestion. We are working to implement and evaluate this baseline now.
>
> **UniRef experiments**
>
>
> > purpose of heldout set:
>
> Thank you for pointing this out. We will revise the paper to better explain this experiment. This is also discussed in the general comment (iii). To respond directly here -- this experiment is looking to see whether an amortization gain can be realized for test sequences that are reasonably different from the train set. The experiment is not trying to get at any biological notion of underlying structural homology. We agree that it could also be interesting to look at generalization across different levels of structural homology; however this is tangential to the focus of the paper, and beyond its scope since it would require pre-training the model with very different datasets constructed to isolate the underlying biological factors of interest.
>
> > short-range
>
> We will add these in the appendix; we wanted to make space for the scatter plot. The trend tracks.
>
>
> >unclear statement:  subsample to M=30
>
> Rephrased to: “During training in each epoch we randomly subsample a different set of 30 sequences from the MSA”. Added M to the 2nd summation of Eq 8.
> This refers only to NPM training. One can think of this as similar to a minibatch: for every gradient update a batch of N random sequences is sampled as input; and N x [M random target sequences] are sampled. Over the course of training, for enough epochs through the dataset, every sequence in the MSA will be sampled.
>
> **Further comments**
>
> > existing methods that currently exists for shallow MSAs. For example, standard semi-supervised methods (...)
>
> We do not want to claim our method is state-of-the-art for contact prediction, it is certainly not. In the paper we use contact precision as a proxy for the underlying accuracy of the Potts model learned by either the amortized objective or the independent model. Our goal is to show an improvement for couplings learned through amortized optimization. Therefore the appropriate comparison is to the independent Potts model. The contact precision simply serves as a way to measure the quality of the model.
> We agree that it would be valuable to include a discussion of other approaches that are relevant for the shallow MSAs and will do this.
>
>
> > Model architecture ablation
>
> Thanks for the suggestion, we are running the architecture ablation experiments on PFAM now and will add these to the paper. We found in preliminary experiments that those architectural improvements (untying, convolutional layers) helped, but did not have time to implement them in the large-scale UniRef runs. We have started new UniRef runs with the improved architecture and expect improved results to be ready to add in the camera-ready version but not by the revision deadline.
>
>
> > please clarify: “we randomly subsample the MSA down to 100 sequences as NPM target sequences”
>
> Thanks for pointing this out, this sentence is out of place. It refers to the same M as in the previous comment; in this case we randomly subsample a different set of 100 sequences from the MSA on each epoch during NPM training.

---

### Author Response · Authors · 2020-11-21
**General comments - response to all reviewers**

We thank the reviewers for their time and helpful feedback. We are pleased to see that all reviewers see value and novelty in our work. (R5: “I find the idea / approach to be very creative(..)  I applaud the authors for taking a unique approach”, R2: “modeling energy landscapes using NLP techniques is a timely and interesting problem”, R4: “idea is particularly compelling”, “sound from a theoretical standpoint (...) [amortized optimization] framework is well motivated (...) “, R3: “approach is novel (...) a well-motivated idea”).

In this comment we address common points in the reviews around (i) need for a more extensive set of experiments, (ii) the relevance of learning Potts models for proteins with low depth MSAs; (iii) the methodology of the UniRef50 experiments.

We would like to emphasize what we see as the main contribution of our work: the introduction of an amortized optimization objective for learning Potts models, and demonstration that the amortization gain expected in theory can be realized. We note all reviewers agree our paper shows this - we believe this is a significant and exciting result convincingly demonstrated by the paper. We hope to foreground this contribution studying amortized optimization, and ask that the experimental evaluations be considered in the context of supporting this result.

We certainly don’t want to claim that the current results present a complete solution to protein structure prediction, rather we propose this approach as a new direction for unsupervised inference of structure from sequences. We further discuss the importance of the low MSA depth problem setting below (where our approach shows an advantage in experiments), and will improve the treatment of this in the revision.

There are many potential avenues to improve the practical utility of the approach which could be explored in the future: e.g. new model architectures or higher capacity models, use of amortized couplings in a supervised pipeline, or combining independent Potts models with amortized couplings. Some of these have been pointed out by reviewers as possible additional experiments. Although it is beyond the scope of the paper to pursue these extensions, we hope the reviewers also see the existence of these and other possibilities for future development as part of the value of the approach.

### (i) Additional experiments we are adding in the revision

We appreciate the suggestions from all reviewers on how to improve the experiments. We outline below an experimental plan for the revision incorporating these suggestions. See also the individual comments for more detail.

* A baseline to Fig 3 (PFAM): Nearest Potts model in train set. Align the validation sequence to all train families from the clan, and use the Potts model from the closest match. (R5)
* An ablation of the architectures (convolutional layers, and multi-head bilinear form weight tied or untied) on PFAM. (R5)
* A plot of the Neural Potts Model objective values against contact precision over the course of training. (R2)
* Additional experiments on HTH and AB_hydrolase clans. (R4, R3).

### (ii) Relevance of the low depth MSA setting

The setting of unsupervised structure learning for low depth MSAs is recognized as an important problem in the literature and is a known limitation of Potts model based methods. Potts models perform poorly when few related sequences are available in an MSA (e.g. Jones et al. 2012, Kamisetty et al. 2013, Moult et al. 2016). This trend is also observed in our experiments (Figures 3 and 4).

The relevance of the low depth setting is reviewed in Tetchner et al. 2014: “many of the largest protein families typically have a structure available for template-based modeling [...] there is clearly more interest in applying covariation analysis to small- and intermediate-sized families.”

Additionally Tetchner et al. 2014 note: “Proteins from higher organisms are a particular problem in terms of available data, as they suffer from there being far fewer sequenced genomes from different species than for bacterial proteins.” Also from Tetchner et al. 2014: “The majority of successes in coevolution-based protein structure prediction have come from proteins that share common ancestry with bacterial proteins, and the comparative lack of available eukaryotic sequences limits the ability to apply covariation methods to families within higher organisms.”

There is further support for the problem setting in the dataset of MSAs we constructed for the UniRef sequences. Figure 6 in the Appendix, which studies the distribution of MSA depths across UniRef50, indicates that 19% of sequences in UniRef50 have MSAs with fewer than 10 sequences, (38% when a minimum query sequence coverage of 80% is specified).

In Figure 4 we find a clear advantage for the Neural Potts model when the MSA has fewer than 10 sequences. Our work shows a path toward improving unsupervised structure inference for low depth MSAs.

---

> ### Author Response · Authors · 2020-11-21
> **General comments continued**
>
> ### (iii) The UniRef50 split
>
>
> R5+R3 asked about the UniRef50 experimental methodology in Section 4.2 and Figure 4. We address the concerns below and will revise the paper to incorporate the feedback from reviewers around this.
>
> This experiment asks whether an amortization gain can be achieved for sequences that are reasonably different from the model’s training data. We believe that using a clustering at 50% identity is reasonable, since this is a method for unsupervised learning from sequences - no structures are used in training the model, and no claims are made about generalization across structural families. We note the baseline independent Potts models have access to the same underlying database of sequences as the Neural Potts model - thus there is no form of data leakage that could put the baseline at a disadvantage.
>
> This experiment does not answer any questions about whether generalization occurs across structural families. While we agree that the question of generalization across structural families (or superfamilies and folds) is certainly of biological interest, a deep investigation of this would be tangential to the main argument of the paper, which is to show that the amortization gain expected from the objective can be realized for sequences outside the training set.
>
> To contextualize the significance of the results on the test set of UniRef50 MSAs, let us consider the setting where the amortized Neural Potts Model (i) matches the independent Potts model on training data: this means the NPM model can predict good quality couplings from a single feedforward pass without access to the full MSA at inference time; (ii) surpasses the independent model on training data: the amortization actually helps NPM to improve over independent Potts models, i.e. it realizes inductive generalization gain; (iii) matches the independent model on unseen sequences: indicates the model is able to synthesize a good Potts model for sequences not in its training data; (iv) surpasses the independent model on unseen data: the model actually improves over an independent Potts model even for sequences not in its training data. In combination these results indicate a non-trivial generalization happens when NPM is trained on UniRef50.
>
> ### References
> * [1] Jones et al. (2012) PSICOV: precise structural contact prediction using sparse inverse covariance estimation on large multiple sequence alignments
> * [2] Kamisetty et al. (2013) Assessing the utility of coevolution-based residue-residue contact predictions in a sequence- and structure-rich era
> * [3] Moult et al. (2016) Critical assessment of methods of protein structure prediction: Progress and new directions in round XI
> * [4] Tetchner et al. (2014) Opportunities and limitations in applying coevolution-derived contacts to protein structure prediction

---

### Author Response · Authors · 2020-11-25
**Uploaded revision**

We have now uploaded a revision, with thanks to all reviewers for help improving the manuscript. The main changes in the new version are:

* Rewrote introduction: discuss importance of low-depth MSA setting [suggested by R5, R4, R3] and remove framing wrt self-supervised learning.
* PFAM Experiments (Sec 4.1 + Appendix):
   - Results on additional clans (NADP_Rossman, HTH, AB_hydrolase); On all 3 of the new clans, NPM matches or exceeds the Independent Potts baseline even on deep MSAs. [suggested by R4, R3]
   - Add a “Nearest Neighbor Potts model” baseline in all plots [suggested by R5]. This baseline is stronger than the independent Potts model for low-depth MSA; but NPM consistently outperforms the new baseline.
   - Add trajectory plot, showing near-monotonic decrease of the amortized pseudo-likelihood loss, and increase of the top-L long range contact precision.  [suggested by R2]
   - Add architecture ablation experiments, showing direct mhbf prediction performance, and showing advantage to adding convolutional layers.  [suggested by R5]
* UniRef50 Experiments (Sec 4.2):
   - Added discussion of the purpose of the training and test set.  [suggested by R5, R3]
   - Added bootstrapped confidence intervals, and more metrics in Appendix. [suggested by R5, R2]
* Added related work section to frame wrt prior work on protein language modeling, unsupervised and supervised contact prediction. [suggested by R4]
* Improved discussion

---

### Decision · Program_Chairs · 2021-01-07
**Final Decision**

**Decision:**

Reject

**Comment:**

This is a creative piece of work wherein learning of what is normally family-specific Potts models is turned into an amortized optimization problem across different families of proteins. The Potts models are learned with a pseudolikelihood approach, and the evaluation of the model against baselines is performed only on a contact prediction problem. This last point is problematic, because on the one hand, the authors use this "as a proxy for the underlying accuracy of the Potts model learned", and on the other hand, claim that "we do not want to claim our method is state-of-the-art for contact prediction, it is certainly not".  Overall, the paper is promising, but is too preliminary on the empirics to warrant publication at this time.